# Particle Engineering of Innovative Nanoemulsion Designs to Modify the Accumulation in Female Sex Organs by Particle Size and Surface Charge

**DOI:** 10.3390/pharmaceutics14020301

**Published:** 2022-01-27

**Authors:** Eike Folker Busmann, Henrike Lucas

**Affiliations:** Faculty of Natural Sciences 1—Biosciences, Department of Pharmaceutical Technology and Biopharmaceutics, Institute of Pharmacy, Martin-Luther-University Halle-Wittenberg, 06120 Halle, Germany; eike.busmann@pharmazie.uni-halle.de

**Keywords:** particle engineering, anionic phospholipid, phosphatidylglycerol, ovarian accumulation, biodistribution, nanoemulsion, surface charge, optical in vivo imaging, cell toxicology

## Abstract

Particle engineering of nanosized drug delivery systems (DDS) can be used as a strategic tool to influence their pharmacokinetics after intravenous (i.v.) application by the targeted adaptation of their particle properties according to the needs at their site of action. This study aimed to investigate particle properties depending on patterns in the biodistribution profile to modify the accumulation in the female sex organs using tailor-made nanoemulsion designs and thereby to either increase therapeutic efficiency for ovarian dysfunctions and diseases or to decrease the side effects caused by unintended accumulation. Through the incorporation of the anionic phospholipid phosphatidylglycerol (PG) into the stabilizing macrogol 15 hydroxystearate (MHS) layer of the nanoemulsions droplets, it was possible to produce tailor-made nanoparticles with tunable particle size between 25 to 150 nm in diameter as well as tunable surface charges between −2 to nearly −30 mV zeta potential using a phase inversion-based process. Three chosen negatively surface-charged nanoemulsions of 50, 100, and 150 nm in diameter showed very low cellular toxicities on 3T3 and NHDF fibroblasts and merely interacted with the blood cells, but instead stayed inert in the plasma. In vivo and ex vivo fluorescence imaging of adult female mice i.v. injected with the negatively surface-charged nanoemulsions revealed a high accumulation depending on their particle size in the reticuloendothelial system (RES), being found in the liver and spleen with a mean portion of the average radiant efficiency (PARE) between 42–52%, or 8–10%, respectively. With increasing particle size, an accumulation in the heart was detected with a mean PARE up to 8%. These three negatively surface-charged nanoemulsions overcame the particle size-dependent accumulation in the female sex organs and accumulated equally with a small mean PARE of 5%, suitable to reduce the side effects caused by unintended accumulation while maintaining different biodistribution profiles. In contrast, previously investigated neutral surface-charged nanoemulsions accumulated with a mean PARE up to 10%, strongly dependent on their particle sizes, which is useful to improve the therapeutic efficacy for ovarian dysfunctions and diseases.

## 1. Introduction

Many approved and marketed drug molecules and even more of those still in development are poorly water-soluble and/or poorly permeable through the physiological barriers, which leads to a low bioavailability at their targeted site of action [1,2,3]. Besides pH modification and salt forms, co-solvency and surfactant solubilization, nanosizing by top-down or bottom-up processes, inclusion complexation, the ongoing development of many different nanosized DDS is a promising approach to enhance the bioavailability of i.v. applied drugs and additionally to reduce side effects [2]. Thereby, particle engineering can be used to create innovative particle designs of such nanosized DDS through the targeted adaptation of their particle properties according to the needs for their intended site of action. In particular, adjusting the particle size, shape, surface charge, and surface flexibility of the nanosized DDS impacts the pharmacokinetics like in vivo blood circulation time, metabolic behavior, or biodistribution profile and consequently the efficiency of the active pharmaceutical ingredient at its targeted site of action [1,4,5].

Nanoparticles with different particle properties were found to accumulate in the tissues of the female reproductive organs [6,7,8,9,10,11,12,13]. On the one hand, this accumulation is desired for the treatment of ovarian dysfunctions or diseases. For ovarian carcinoma therapy, i.v. delivered nanosized DDS loaded with chemotherapeutic agents sometimes combined with contrasting or fluorescent agents are used for therapeutic, diagnostic, or even theranostic approaches [14,15,16,17,18], as well as for the therapy of primary ovarian insufficiency with a constant parenteral low dose hormonal infusion mimicking the daily hormonal ovarian production [19,20]. On the other hand, unintended accumulation of i.v. delivered nanoparticles in the sex organs were reported as leading to various toxicity risks like ovarian inflammation, congestion, fibrosis, extravasations of red blood cells, apoptosis in ovarian cells, or the decrease in the number of follicles [6,9,10].

By the incorporation of (highly) lipophilic active ingredients into the liquid lipid matrix, nanoemulsions are already used as nanosized DDS exhibiting numerous advantages: accepted regulatory status, since many ingredients are already used for parenteral nutrition and drug emulsions; usually low toxicity using physiological and readily biodegradable substances like triglycerides and phospholipids; easy processes obtaining very high loads of the active ingredients by pre-loading during or post-loading after manufacturing of the nanoemulsions; high protective potential against hydrolysis or oxidation of the active ingredient by its complete enclosure inside the lipid matrix showing only a very low active ingredient transfer out of its core or to the droplet surface; plus providing a water-based formulation with low viscosity suitable for parenteral i.v. applications [5,21,22].

Our previous studies revealed easy to manufacture nanoemulsions using a phase inversion-based process with tunable particle sizes within a range of 16 to 175 nm and an almost neutral surface charge [23], whose composition is illustrated in Figure 1a. In vivo studies revealed a particle size as well as age-dependent ovarian accumulation after their i.v. injection. Prepubescent juvenile mice showed nearly no ovarian accumulation. But after puberty, the ovarian accumulation increased in fertile adult mice, depending on the particle size. Thereby, increasing the nanoemulsion particle size from 25 to 150 nm in diameter doubled the measured relative fluorescence intensity in the ovaries. However the relative fluorescence intensity in the ovaries decreased again with reproductive aging and the accompanying declining fertility of senescent mice [24].

This study investigates whether the accumulation in the female reproductive organs of i.v. injected nanoemulsions can be controlled by adapting the nanoemulsion particle design. Therefore, the surface of the previously investigated neutral surface-charged nanoemulsions was modified by the incorporation of PG into the stabilizing MHS layer to create a negatively charged surface of the nanoemulsion droplets, as shown in Figure 1b. The phospholipid PG possesses an anionic charge in its hydrophilic head group, which contains a second glycerol moiety [25]. It is a natural constituent in mammals and occurs as a lung surfactant as well in the plasma [25,26,27]. Furthermore, PG was reported as non-toxic and non-hemolytic, plus it revealed anti-inflammatory activity in vitro [28,29,30] and thus seemed suitable to formulate nanoemulsions with a negative surface charge.

For the comparison of these negatively surface-charged nanoemulsions with the previously investigated neutral surface-charged nanoemulsions, the medium-chain triglyceride (MCT) core was loaded as well with the near-infrared fluorescent dye DiR (DiLC18(7)), which was already deployed for in vivo fluorescence imaging and reported as non-toxic [7,23,24,31,32,33]. Using such a near-infrared fluorescent dye allowed for the minimizing of the autofluorescence signal of the mice plus probing in larger tissue depths than a few hundred microns within ultraviolet and visible wavelengths because of the light absorption of the tissue cell components [34,35]. Additionally, the immunocompetent and hairless SKH1-*Hr^hr^* mouse strain was chosen, which loses its hair within three weeks after birth and thus excludes possible interferences between fur and both excitation as well as emission light [35,36,37]. The dialkylcarbocyanine DiR with a reported logP of 17.4 is highly lipophilic and insoluble in water [38]. Thus the DiR stayed nearly completely inert in the MCT core of the previously investigated neutral surface-charged nanoemulsions so that depending on the particle size, only a very small amount of maximal 1.3% DiR was transferred to blood cells as an acceptor [24]. Since DiR was shown to degrade to fluorescent molecules with different properties at pH values below 4, possible changes of the fluorescence signal with DDS developing an acidic microenvironment should be considered in its data evaluation [39].

## 2. Materials and Methods

Kollisolv^®^ MCT 70 (medium-chain triglyceride, MCT) and Kolliphor^®^ HS 15 (macrogol 15 hydroxystearate, MHS) were kindly provided by BASF SE (Ludwigshafen, Germany), and LIPOID EPG (egg phosphatidylglycerol sodium salt, PG) by Lipoid GmbH (Ludwigshafen, Germany). Sodium chloride and the near-infrared fluorescent dye DiR (1,1′-dioctadecyl-3,3,3′,3′-tetramethylindotricarbocyanine iodide) were purchased from Grüssing GmbH (Filsum, Germany) or Invitrogen/Thermo Fisher Scientific Inc. (Carlsbad, CA, USA), respectively. Dulbecco’s modified eagle medium—high glucose with 4500 mg/L glucose, L-glutamine, sodium bicarbonate, with and without sodium pyruvate (DMEM w/NaP or DMEM w/o NaP, respectively), fetal calf serum (FCS), penicillin-streptomycin (P/S), Triton^TM^ X-100, fluorescent dye resazurin sodium salt, sodium citrate dehydrate and Dulbecco’s phosphate buffered saline 10x (PBS) were purchased from Sigma-Aldrich Chemie GmbH (Steinheim, Germany). Double distilled and 0.2 µm sterile-filtered water was used in all experiments and analytics.

### 2.1. Preparation of Isotonic Nanoemulsions with Negative Surface Charge

For the preparation of isotonic nanoemulsions with a negative surface charge, a previously developed phase-inversion-based process [23] was modified by incorporating the anionic phospholipid PG into the stabilizing MHS layer. For the parameter screening, varying PG and MHS mass shares were dispersed thoroughly with 30 wt.% MCT in an aqueous NaCl solution under magnetic stirring at ~750 rpm. The emulsion was heated to 99 °C undergoing a phase inversion from an o/w to a w/o emulsion. It was cooled back into its phase inversion zone and shock diluted with ice-cold water.

A previously developed equation system for the production of isotonic nanoemulsions with a neutral surface charge [23] was modified to calculate the amounts of the respective educts, including the incorporated PG to achieve the negative surface charge. Thereby, the mass MCT $m_{MCT}$ is the product of the desired total mass nanoemulsion $m_{tot}$ and the desired mass share MCT in the final product $x_{MCT,p}$ after the shock dilution, which was set to 8 wt.% MCT:(1)$$m_{MCT}=m_{tot}\times x_{MCT,p}$$

The mass of the stabilizing anionic surfactant MHS $m_{MHS}$ is calculated by multiplying the mass MCT with the ratio of the mass shares of MHS $x_{MHS,0}$ and MCT $x_{MCT,0}$ before the shock dilution. Thereby, $x_{MHS,0}$ was varied between 10 to 40 wt.% MHS and $x_{MCT,0}$ was set to 30 wt.% MCT:(2)$$m_{MHS}=m_{MCT}\times \frac{x_{MHS,0}}{x_{MCT,0}}=m_{tot}\times x_{MCT,p}\times \frac{x_{MHS,0}}{x_{MCT,0}}$$

The mass of the incorporated anionic phospholipid PG $m_{PG}$ is the product of the mass MCT with the ratio of the mass shares of PG $x_{PG,0}$ and MCT $x_{MCT,0}$ before the shock dilution, where $x_{PG,0}$ was varied between 0 to 6 wt.% PG:(3)$$m_{PG}=m_{MCT}\times \frac{x_{PG,0}}{x_{MCT,0}}=m_{tot}*x_{MCT,p}\times \frac{x_{PG,0}}{x_{MCT,0}}$$

The mass of the aqueous NaCl solution $m_{NaCl-Sol}$ is similarly calculated by multiplying the mass MCT with the ratio of the mass shares of NaCl solution $x_{NaCl-Sol,0}$ and MCT $x_{MCT,0}$ before the shock dilution, where $x_{NaCl-Sol,0}$ is the difference of 1 with the mass shares of the other three components MCT, MHS, and PG:(4)$$m_{NaCl-Sol}=m_{MCT}\times \frac{x_{NaCl-Sol,0}}{x_{MCT,0}}=m_{tot}\times x_{MCT,p}\times \frac{1-x_{MCT,0}-x_{MHS,0}-x_{PG,0}}{x_{MCT,0}}$$

The mass of ice water $m_{ice\;water}$ for the shock dilution is the difference of the total mass and all previously calculated masses of MCT, MHS, PG, and NaCl solution:(5)$$m_{ice\;water}=m_{tot}-m_{MCT}-m_{MHS}-m_{PG}-m_{NaCl-Sol}=m_{tot}\times \left(1-\frac{x_{MCT,p}}{x_{MCT,0}}\right)$$

To achieve a similar osmolality of blood $O_{blood}$ at 300 mosmol/kg, the calculation of the necessary salinity of the aqueous NaCl solution before shock dilution [23] was edited as follows:(6)$$c_{NaCl,0}=\;\left(O_{blood}-O_{MHS}\left(c_{MHS,p}\right)-O_{PG}\left(c_{PG,p}\right)\right)\times \frac{M_{NaCl}}{\phi_{NaCl}\times n_{NaCl}}\times \frac{m_{NaCl-Sol}+m_{ice\;water}}{m_{NaCl-Sol}}$$

As shown in the Appendix A, PG did not influence the osmolality, so that the term $O_{PG}\left(c_{PG,p}\right)$ was set to zero. The term $\frac{M_{NaCl}}{\phi_{NaCl}*n_{NaCl}}$ was previously determined as 2.922 × 10^−3^ kg/mmol through the reciprocal slope of the fitted linear van’t Hoff law, using ideal osmolality data of aqueous NaCl solutions after Ph. Eur. 2.2.35. Experimental data revealed a well corresponding fit with the modified polynomial van’t Hoff equation of 4th degree to calculate the term $O_{MHS}\left(c_{MHS,p}\right)$ [23]:(7)$$\begin{array}{c} O_{MHS}\left(c_{MHS,p}\right)\;=4.6293\;\frac{mmol}{kg}\times c_{MHS,p}+0.6041\;\frac{mmol}{kg}\times c_{MHS,p}^{2}-0.0235\;\frac{mmol}{kg}\times c_{MHS,p}^{3}\\ \;\;\;\;\;\;\;+\;0.0008\;\frac{mmol}{kg}\times c_{MHS,p}^{4} \end{array}$$

For the animal trials, three isotonic and negatively surface-charged nanoemulsions were chosen with targeted particle sizes of 50 (negative NE50), 100 (negative NE100), and 150 nm (negative NE150), whose compositions are listed in Table 1. Their educts were sterilized before their preparation. MHS and PG mixed in aqueous NaCl solution as well as water for the shock dilution were autoclaved for 30 min at 121 °C and 2 bar. MCT was sterilized by dry heat for 30 min at 180 °C. The nanoemulsion preparations were conducted one day before every animal experiment and under aseptic conditions in a laminar flow bench. Ethanol of the DiR stock solution was evaporated and re-dissolved in the sterilized MCT at a concentration of 0.1 mg/g for fluorescent labeling of the nanoemulsions. The further preparation of these nanoemulsions was done as described previously, but under aseptic conditions and finally sterile filtered with a 0.2 µm polyethersulfone (PES) filter into sterilized vials.

### 2.2. Particle Size and Zeta Potential Analysis

Both particle sizes and zeta potentials were determined with the Malvern Instruments Zetasizer Nano ZS by Malvern Panalytical GmbH (Kassel, Germany). For particle size measurements, samples were diluted at 1:100 in water and measured in triplicate with 15 runs each at 25 °C in backscattering mode. The zeta potential was determined in triplicate by diluting each sample 1:10 in 0.1x PBS with pH 7.4 at 25 °C with 10 to 50 runs per measurement.

### 2.3. Osmolality and pH Measurements

The osmolality of each sample was determined in triplicate with the Semi-Mikro Osmometer by KNAUER Wissenschafliche Geräte GmbH (Berlin, Germany). The pH values of each sample were determined by single-point measurements with the Portamess^®^ 913 pH meter by Knick Elektronische Messgeräte GmbH (Berlin, Germany).

### 2.4. Longterm Stability

According to the ICH guidelines and Q1A recommended conditions, the samples were stored at 5 ± 3 °C, room temperature, and 40 ± 2 °C. Particle size analysis of the samples was conducted directly after production and then after two, four, eight, and 12 weeks of storage.

### 2.5. Cellular Toxicity

Cellular toxicity assays of the three negatively surface-charged nanoemulsions were carried out in triplicate at eight replicates per run on normal human dermal fibroblasts (NHDF) and mouse embryonic fibroblasts (3T3) obtained from Promocell (Heidelberg, Germany) or ATCC (Manassas, VA, USA), respectively. Therefore, 20,000 NHDF cells or 10,000 3T3 cells were seeded in 11 of the 12 columns of 96-well plates in 100 µL DMEM *w*/*o* NaP for the NHDF cells of DMEM w/NaP for the 3T3 cells, both media with an additional 10 vol.% FCS and 1 vol.% P/S. The cells were grown for 24 h at 37 °C and 5% CO_2_. The first of the 12 columns was used for the determination of the blank filled solely with 200 µL corresponding cell culture medium without any cells, the second column with cells was left untreated by adding 100 µL cell culture medium as a negative control for viable cells and the third column was treated by adding 100 µL of 0.05 vol.% Triton^TM^ X-100 solution (final concentration 0.025 vol.%) as a positive control of fully inhibited cells. The cells in the remaining nine columns were treated with different dilutions of the investigated nanoemulsion. Therefore, the nanoemulsion was 0.2 µm sterile filtered and diluted up to 10 times at ratios of 1:2.35, with the corresponding cell culture medium and 100 µL of the diluted nanoemulsion added to the NHDF or 3T3 cells. The cells were then incubated for 24 h. By using a resazurin reduction assay, the cell viability was determined by adding 20 µL of 440 µM resazurin solution. The added resazurin solution was mixed thoroughly with the cell culture medium by carefully withdrawing the medium back and forth into the pipette tips and incubated for another 2 h. The fluorescence intensity was then determined with the BioTek^®^ Instruments Cytation^TM^ 5 Cell Imaging Multi-Mode Reader by Agilent Technologies Inc. (Santa Clara, CA, USA) using the RFP 531(excitation)/593(emission) filter set. The cell viability was calculated as the percentage of the negative controls (untreated cells) after subtraction of the blank (only medium without cells). Using linear interpolation, the median inhibitory concentration IC50 was determined. Additionally, to the fluorescence measurements, the morphology of the incubated 3T3 or NHDF fibroblasts was observed by phase-contrast microscopy at 20× magnification.

### 2.6. Investigation of the Interaction with Blood Cells

By adding 0.109 M sodium citrate solution at a ratio of 1:9 to the withdrawn whole blood of untreated mice, the whole blood was stabilized against coagulation. 50 µL nanoemulsion was added to 100 µL stabilized whole blood, mixed thoroughly, and incubated for 4 h at room temperature. The blood cell fraction was then separated from the plasma by centrifugation at 3000 rpm for 15 min with the MiniSpin^®^ Centrifuge by Eppendorf SE (Hamburg, Germany). The plasma was pipetted off and the pellet with the blood cell fraction was washed by repeatedly redispersing in 1x PBS and centrifuging three times in total. Both plasma and blood cell fractions as well as untreated whole blood as blank were transferred into a 96-well plate. The fluorescence intensity of the samples was determined as average radiant efficiency (ARE) by fluorescence imaging as described in Section 2.8. The portion of the average radiant efficiency (PARE) of both fractions was calculated after subtraction of the untreated whole blood ARE, according to Section 2.9 and Equation (8).

### 2.7. Animal Handling

All in vivo protocols complied with the guidelines of the Federation for Laboratory Animal Science Associations (FELASA) and were approved under file number 42502-2-1456 MLU by the local authorities of Saxony-Anhalt, Germany. The in vivo studies were performed in adult (age: 15–30 weeks, mean body weight 29.3 ± 1.5 g) female hairless immunocompetent SKH1-*Hr^hr^* Elite mice (mouse model of Charles River, Sulzfeld, Germany) bred by the center for basic medical research (ZMG) of the Martin Luther University, Halle-Wittenberg, Germany. The mice were housed in windowless and fully automated air-conditioned rooms with barrier-maintained mouse colonies in closed cage systems type II long with a maximum of five mice per cage according to the EG-guideline 86/609/EWG and GV-SOLAS guidelines. The light and temperature regime was controlled automatically and the feeding followed the special diet for lab mice, having access to water and food ad libitum.

The experimental procedure of the animal trials is illustrated in Figure 2. Using a 30 G cannula and a restrainer darkened with red foil, five mice for each nanoemulsion formulation were slowly i.v. injected in the tail veins with 200 µL nanoemulsion, which results in a mean dose of 6.8 ± 0.3 µL/g body weight. For in vivo fluorescence imaging directly after the bleeding of the injection site stopped (5–10 min after application), as well as after 24 h, the mice were anesthetized with the XGI-8 Gas Anesthesia System by Caliper Life Sciences (Waltham, MA, USA) and Matrx VIP 3000^®^ Isoflurane Vaporizer by Midmark Corporation (Dayton, OH, USA) at a 1–2% isoflurane-oxygen mixture at ~3.0 L/min O_2_ initial flow in the induction chamber and ~1.5 L/min steady flow during fluorescence imaging, both flows at 21.1 °C and atmospheric pressure. During fluorescence imaging, the mice were placed on a 37 °C temperature-controlled stage, preventing the body temperature from decreasing. The mice were sacrificed by cervical dislocation 24 h after injection. For further ex vivo analysis, 17 organs were excised and whole blood samples were withdrawn during necropsy.

### 2.8. Fluorescence Imaging

In vivo and ex vivo fluorescence imaging was performed with the IVIS^®^ SPECTRUM fluorescence imager by PerkinElmer Inc. (Waltham, MA, USA) using the epi-illumination mode at auto exposure with the following filter pair: excitation at 745 nm with 20 nm bandwidth, emission at 800 nm with 30 nm bandwidth. For the in vivo fluorescence imaging of every single mouse, the field of view was set to D using the mouse as the set subject. The field of view was set to C using the well plate as the set subject for the ex vivo fluorescence imaging of the excised organs displayed in a 12-well plate.

### 2.9. Image and Data Processing of the Fluorescence Images

PerkinElmer^®^ Living Image^®^ 4.7.3 Software was used for the image and data processing as described previously [24]. For the adjustment of the fluorescence images, brightness was set to 100, the contrast to 1.5, binning to 2, opacity to 60, and the smoothing to “none”, using the “rainbow2” color table in reverse and logarithmic scale. The color scale of the radiant efficiency was set to 3.0 × 10^7^ (p/sec/cm^2^/sr)/(µW/cm^2^) minimumto cut off the autofluorescence of mouse tissue, which was determined beforehand with untreated mice as blank. The maximum was set to 3.0 × 10^9^ and 5.0 × 10^9^ (p/sec/cm^2^/sr)/(µW/cm^2^) above the maximal detected radiant efficiency of all taken in vivo or ex vivo fluorescence images, respectively.

For data processing of the ex vivo fluorescence images, the fluorescence intensity of the excised organs was determined as ARE by drawing a region of interest (ROI) automatically around the outlines of each organ with the “auto1” function, and if not applicable manually, with the “free draw” function. For comparison, the PARE of each single organ n of a mouse m $PARE_{m,n}$ was expressed as the portion of the sum of all measured ARE-signals of the 17 excised organs plus the withdrawn blood (*n* = 1…18) of the mouse m $ARE_{m,n}$:(8)$$PARE_{m,n}=\frac{ARE_{m,n}}{\sum_{n=1}^{18}ARE_{m,n}}$$

The mean PARE $\emptyset PARE_{n}$ was calculated for the 3D bar chart and logarithmic contour plots for the discussion of the biodistribution in Section 3.7:(9)$$\emptyset PARE_{n}=\frac{\sum_{m=1}^{5}PARE_{m,n}}{5}$$

The mouse with the smallest deviation $Dev_{m}$ from the arithmetic mean of all calculated $PARE_{m,n}$ within the group of the five mice was then chosen as a representative mouse for the in vivo and ex vivo fluorescence images in the Section 3.6 and Section 3.7, according to the following equation:(10)$$Dev_{m}=\sum_{n=1}^{18}\left|\emptyset PARE_{n}-PARE_{m,n}\right|$$

Using the software IBM^®^ SPSS^®^ Statistics Version 24, statistical analysis was performed by pairwise Mann-Whitney-U-Tests (MWU) to investigate the effect of the particle size on the accumulation within both groups’ negatively and neutral surface-charged nanoemulsions. To investigate the effect of the surface charge on the accumulation, pairwise MWUs were performed between the pairs of similar-sized nanoemulsions with a negative and neutral surface charge. MWU is a robust test for the statistical analysis of small samples and does not presuppose a normal distribution, suitable for such a small number of samples with only five mice per administered nanoemulsion formulation.

## 3. Results and Discussion

### 3.1. Screening of the Nanoemulsion Formation with Negative Surface Charge

Screening trials were conducted to achieve nanoemulsions with a negative surface charge and targeted particle sizes of 50, 100, and 150 nm in diameter, with a narrow size distribution. Furthermore, it was aimed to achieve isotonicity and thus reduce the risk of crenation or shriveling of the blood cells as well as pain at the site of application. The contour plots in Figure 3 show the resulting (a) particle sizes, (b) polydispersity index (PDI), (c) zeta potential, and (d) osmolality of the produced nanoemulsions at different MHS and PG mass shares at a constant MCT mass share of 8 wt.%. Additionally, the Appendix A shows the same experimental data as scatter plots against the PG:MHS ratio.

The hydrodynamic particle diameters were measured by dynamic light scattering and were expressed as the commonly used Z average (z_ave_) and the width of the overall particle size distribution as the common dimensionless PDI (ISO 22412 and [40]). Depending on the mass shares of the non-ionic surfactant MHS and the anionic phospholipid PG, nanoemulsions were formed with particles of 27 to 148 nm in diameter (Figure 3a) and narrow size distributions with PDIs of 0.02 to 0.15 (Figure 3b). Thereby, the increase of both surfactant mass shares MHS and PG led to consistently decreasing particle sizes of the nanoemulsions. In the case of the PDIs, the composition passed through an optimum, where at first the PDIs decreased with increasing surfactant mass shares from 0.15 to very narrow distributions with PDIs below 0.05 accompanying the decreasing particle sizes of the nanoemulsions. Thereby, the lowest PDIs were detected for the nanoemulsions with particle sizes around 60 nm. With the further increase of both surfactant mass shares, the PDIs increased slightly up to 0.08, although with the further decreasing particle sizes of the nanoemulsions.

By electrophoretic light scattering in 0.1× PBS at physiological pH 7.4, the electrokinetic potentials at the slipping plane of the nanoemulsion droplets were determined, and these results are plotted in Figure 3c. With the increasing mass share of the anionic phospholipid, the zeta potential decreased from nearly neutral with −2 mV at 0.00 wt.% PG up to nearly −30 mV at 1.60 wt.% PG and 2.67 wt.% MHS. Thereby, the resulting zeta potential depended only on the PG:MHS ratio of the nanoemulsion composition, as shown in the Appendix A. Hence, increasing this ratio and therewith increasing the portion of PG on the surface of the nanoemulsion droplets decreased the zeta potential of the nanoemulsions.

Figure 3d shows the resulting osmolalities of the screened nanoemulsions, which were determined by cryoscopy. All produced nanoemulsions were nearly isotonic at osmolalities between 283 and 318 mosmol/kg.

### 3.2. Physicochemical Properties of the Three Chosen Negatively Surface-Charged Nanoemulsions

For further in vitro and in vivo investigation, three compositions were chosen with the targeted particle sizes of 50, 100, and 150 nm, and the most possible negative zeta potentials. Thereby, these three compositions had the lowest investigated MHS mass share of 2.67 wt.%, since this non-ionic surfactant was determined as the main potentially toxic ingredient of previously investigated isotonic and neutral surface-charged nanoemulsions [23]. Negatively surface-charged nanoemulsions of different particle sizes were reproducible and formed by only adjusting the PG mass shares to 1.60 wt.% for the negative NE50 (red bars), 0.80 wt.% PG for the negative NE100 (blue bars), and 0.27 wt.% PG for the negative NE150 (green bars), as shown with their resulting physicochemical properties in Figure 4. By adjusting only the PG mass shares to 1.60 wt.% for the negative NE50 (red bars), to 0.80 wt.% for the negative NE100 (green bars), and to 0.27 wt.% for the negative NE150 (green bars), the three targeted nanoemulsion formulations were reproducibly formed with the in Figure 4 depicted physico-chemical properties.

The three negatively surface-charged nanoemulsions had particle sizes very close to their targeted particle sizes with the negative NE50 at 51.0 nm, the negative NE100 at 95.7 nm, and the negative NE150 at 147.5 nm in diameter. With PDIs below 0.05, both negative NE50 and negative NE100 had very narrow monomodal size distributions, as shown in the Appendix A. The PDI of 0.148 indicated as well a narrow monomodal size distribution for the negative NE150 (Appendix A). The zeta potentials of these three nanoemulsions were determined at −28.8 mV for the negative NE50, −26.5 mV for the negative NE100, and −20.2 mV for the negative NE150. All three nanoemulsion compositions were isotonic with osmolalities between 305.0 to 307.2 mosmol/kg, thus ensuring a painless i.v. application without any vascular damage [41,42]. The pH values between 4.6 and 5.0 indicated a slight acidity of the three nanoemulsions but were still within the acceptable range of pH 3–11 for small volume i.v. injections [41,43]. Within the short time between the aseptic nanoemulsion preparation one day before i.v. application until the end of the animal trials, acid-induced degradation of the incorporated fluorescent dye DiR for the in vivo fluorescence imaging was very unlikely, since DiR was described as chemically stable for pH values above 4.0 [39].

### 3.3. Longterm Stability

As metastable systems, nanoemulsions are prone to physical destabilization through flocculation, coalescence, Ostwald ripening, or even creaming at various timescales, which may take up to many months or years [44,45]. Therefore, the three negatively surface-charged nanoemulsions were investigated against possible physical destabilization according to the ICH guidelines of Q1A at 5 ± 3 °C, room temperature, and 40 ± 2 °C. Figure 5 shows their changes in particle size and PDI over 12 weeks at the three recommended storage conditions.

At both storing conditions of 5 ± 3 °C and room temperature, all three negatively surface-charged nanoemulsions showed sufficient stability of their particle size over the complete time of 12 weeks. Their intensity weighed particle size distributions stayed monomodal and still narrowly distributed, as shown in the Appendix A. Exposing the negatively surface-charged nanoemulsions to increased thermal stress of 40 ± 2 °C, led to a small increase of the particle size of the negative NE50 after four weeks of storage, which further increased up to 587 nm after 12 weeks of storage. The intensity weighed particle size distribution became bimodal and broadly distributed, as shown in the Appendix A. While the particle size of the negative NE100 increased slightly after 12 weeks of storage at 40 ± 2 °C with a shift of the intensity weighed size distribution to bigger particle sizes (Appendix A), the particle size of the negative NE150 stayed stable throughout the complete investigated time with a remaining narrow intensity weighed particle size distribution (Appendix A).

### 3.4. Cellular Toxicity

The cellular toxicity of the negatively surface-charged nanoemulsion was determined by resazurin reduction assays with 3T3 as well as NHDF fibroblasts after 24 h of incubation. The resulting dose-response curves of the cell viability against the concentration of the three nanoemulsions are shown in Figure 6.

With increasing nanoemulsion concentration, all three nanoemulsions revealed growth-promoting effects. Thereby, the negative NE50 had its maximal cell viability at a nanoemulsion concentration of 7.0 mg/mL for both fibroblasts lines. The negative NE100 and negative NE150 had both their maximal cell viabilities at 16.4 mg/mL for both fibroblasts lines. With increasing concentrations beyond their maxima, cytotoxic effects became dominant, leading to a rapid decrease in cell viability for the negative NE100 and NE150 for both fibroblast lines and less rapid for the negative NE50. This less rapid decrease in cell viability for the negative NE50 might derive from a more robust surfactant layer by higher electrostatic forces through the high amount of incorporated anionic phospholipid PG in the stabilizing layer of the negative NE50 nanoemulsion droplets. The cytotoxic effects of the three negatively surface-charged nanoemulsions were likely induced by the non-ionic surfactant MHS, which was detected in previous research as the main cell viability inhibiting ingredient on various cell lines [23,46,47,48]. Thereby, a formation of needle-like 12 hydroxystearic acid crystals as a metabolic degradation product of MHS likely caused cell death in vitro [23]. A similar formation of the needle-like 12 hydroxystearic acid crystal formation was observed for the negatively surface-charged nanoemulsions, as shown in the Appendix A.

However, such a crystal formation is unlikely to occur in vivo because of different transport and metabolic conditions. Instead, 12 hydroxystearic acid was found to be deposited in the body lipids, mainly in the abdominal fat, after feeding rats with a 12 hydroxystearic acid-containing diet for up to 16 weeks. Furthermore, the 12 hydroxystearic acid was degraded to its metabolites 10 hydroxypalmitic acid and 8 hydroxymyristic acid by successive losses of two carbon units at the carboxyl end of the fatty acid chain [49]. Another in vivo study showed a rapid breakdown of the 12 hydroxystearic acid, where only one-fifth of the radioactively labeled 12 hydroxystaeric acid could be recovered 5 min after i.v. injection from the whole rat [50].

Since the dose-response curves of the cell viabilities did not show a sigmoidal curve, the median inhibitory concentrations IC50 of the three nanoemulsions were determined by linear interpolation between the two data points above and below 50% cell viability. The calculated IC50 values, at which the fibroblasts possess half the viability of untreated cells, are displayed in Figure 7 for both fibroblast lines.

After 24 h incubation, both negative NE100 and negative NE150 had similar IC50 values at concentrations of 56.0 and 60.4 mg/mL (3T3 fibroblasts) as well as 80.1 and 75.0 mg/mL (NHDF fibroblasts), respectively. The negative NE50 had substantially higher IC50 values at 103.4 mg/mL (3T3 fibroblasts) and 195.8 mg/mL (NHDF fibroblasts), resulting from the previously discussed less rapid decrease of both 3T3 and NHDF dose-response curves in comparison to the 3T3 and NHDF dose-response curves with the negative NE100 and negative NE150. In comparison, these three negatively surface-charged nanoemulsions had substantially higher IC50 values than previously investigated neutral surface-charged MCT nanoemulsions with equivalent particle sizes. Their IC50 values after 24 h incubation were determined to be between 14.6 to 32.3 mg/mL (3T3 fibroblasts) or 23.9 to 51.5 mg/mL (NHDF fibroblasts), depending on their particle size and consequently on their MHS content [24]. The almost doubled IC50 values of these three nanoemulsions were derived by reducing the potentially cytotoxic MHS concentrations from 4.0–8.8 wt.% for the neutral surface-charged nanoemulsions (depending on the targeted particle size) down to 2.67 wt.% for the negatively surface-charged nanoemulsions for all three targeted particle sizes. Hence, the incorporation of the anionic phospholipid PG into the stabilizing layer of the MCT nanoemulsion droplets decreased the cytotoxicity substantially in comparison to the neutral surface-charged MCT nanoemulsions. Therefore, the safety for a complication-free i.v. application was increased by maintaining equivalent targeted particle sizes.

### 3.5. Blood Cell Interaction

The interaction of blood cells with the three negatively surface-charged nanoemulsions loaded with the fluorescent dye DiR was investigated by incubating a volume of 50 µL of every nanoemulsion in 100 µL sodium citrate stabilized whole mouse blood for 4 h at room temperature. The portions of the detected fluorescent signal of both separated plasma and blood fractions were calculated as PARE and are displayed in Figure 8. The PARE of the blood cell fraction slightly increased from 0.2% to 0.9% with decreasing particle size. This derived either by a marginal cellular uptake of the smaller sized nanoemulsion droplets into the blood cells or by a very small amount of transferred DiR molecules from the lipid core of the nanoemulsion droplets through the blood cell membranes because of a highly increased specific surface area available for such a transfer by downsizing the nanoemulsion droplets. Overall, the very low PARE of the blood cell fraction indicated that all three nanoemulsions merely interacted with the blood cells and that for the most part the highly lipophilic fluorescent dye DiR with a reported logP of 17.4 [38] stayed inert in the MCT core of the negatively surface-charged nanoemulsions, similar to previously investigated neutral surface-charged nanoemulsions of the same particle size range [24].

### 3.6. In Vivo Fluorescence Imaging

The incorporation of the near-infrared fluorescent dye DiR into the lipid core enabled the detection of accumulating nanoemulsions in larger tissue depths, avoiding the light adsorption of tissue cell components in ultraviolet and visible wavelengths [34]. The accumulation of the three negatively surface-charged nanoemulsions in adult mice (age: 15–30 weeks) was monitored in vivo by noninvasive fluorescence imaging directly after the i.v. application into the tail vein (5–10 min), and again 24 h thereafter. Figure 9 shows the lateral, ventral, and dorsal in vivo fluorescence images of the representative mice (one for each nanoemulsion according to Equation (10)). For better comparability of the biodistribution profiles affected by the nanoemulsion particle design, the results of the animal trials with the previously developed neutral surface-charged nanoemulsions i.v. injected in adult SKH1-*Hr^hr^* Elite mice (age: 12–39 weeks) are included in the following figures. Their particle properties were reported with particle sizes of 25.7 nm (neutral NE25), 50.5 nm (neutral NE50), 97.7 nm (neutral NE100), and 144.9 nm (neutral NE150) at zeta potentials between −2 to −3 mV at pH 7.4 [24]. Thereby, their MCT concentration of 8.0 wt.% as well as their DiR-load of 0.1 mg/g were exactly similar to the MCT concentration and DiR load of the negatively surface-charged nanoemulsions and therefore suitable for a direct comparison by fluorescence imaging.

The previously investigated neutral surface-charged nanoemulsions were reported to accumulate in the liver quickly after their i.v. application [24], which was observed as well for all three negatively surface-charged nanoemulsions in the ventral view, as shown by the white arrows. This indicated a rapid hepatic uptake of both types of nanoemulsions into the RES, involving the Kupffer cell-mediated phagocytosis in the liver similar to a variety of other nanoparticulate formulations observed within a few minutes to 1.5 h after parenteral application [8,51,52,53,54]. The intensity of the fluorescence signal, stated as radiant efficiency, increased for the liver within 24 h after the i.v. application. Thereby, the radiant efficiency increased moderately for all three negatively surface-charged nanoemulsions, contrary to the previously investigated neutral surface-charged nanoemulsions, where the highest radiant efficiencies in the liver were detected, especially for the small-sized formulations NE25 and NE50 [24]. This indicated that the incorporation of the anionic phospholipid PG into the stabilizing layer and hence the targeted decrease of zeta potential from nearly neutral down to at least −20.2 mV might lead to a more wide biodistribution of the negatively surface-charged nanoemulsions across other organs, which was observed as well for negatively surface-charged PG liposomes in comparison to neutral surface-charged phosphatidylserine liposomes [54]. Despite the hepatic accumulation, ovarian accumulation was detected 24 h after i.v. application of the three negatively surface-charged nanoemulsions in the adult mice. Depending on the reproductive aging of the mice, this phenomenon was as well detected in all four previously investigated neutral surface-charged nanoemulsions. Thereby, nearly no ovarian accumulation was detected before the onset of puberty, peaking at its maximum during the most fertile lifetime of adulthood, and then decreasing again with increasing senescence of the mice [24].

### 3.7. Ex Vivo Fluorescence Imaging and Biodistribution of the Nanoemulsions

For ex vivo analysis, the mice were sacrificed 24 h after their i.v. injection. Seventeen organs were excised and whole blood was withdrawn by heart puncture during necropsy. Ex vivo fluorescence imaging was conducted of the excised organs and the withdrawn whole blood. Around the outlines of every single organ, a ROI was drawn using the PerkinElmer^®^ Living Image^®^ 4.7.3 Software to determine the fluorescence intensity as ARE. The PARE of every single organ was calculated for each mouse according to Equation (8), followed by the mean PAREs for the five mice per nanoemulsions formulation according to Equation (9), which are provided in the Appendix A. Because of the small number of samples of five mice per i.v. injected nanoemulsion formulation, statistical analysis was performed with MWU. The results are provided in Appendix A. Figure 10 gives an overview of all calculated mean PAREs of the three negatively surface-charged nanoemulsions and, additionally, for better comparability, the four previously investigated neutral surface-charged nanoemulsions [24]. Thereby, the mean PARE and the individual PARE values of every single mouse are plotted as a combined 3D bar chart and 3D scatter chart, respectively. Depending on their targeted particle size, the nanoemulsion formulations NE25 (orange bars and scatter), NE50 (red), NE100 (blue), and NE150 (green) are plotted according to their negative or neutral zeta potential along the *x*-axis. The ex vivo excised organs are plotted against the *y*-axis, while the mean and individual PARE values of the mice are plotted along the *z*-axis.

Additional to the overview of all excised organs, the mean PARE of the seven organs with the highest accumulations (liver, spleen, uterus + ovaries, heart, stomach, kidneys, and caecum) are plotted as contour plots against the particle size of both negatively and neutral surface-charged nanoemulsion formulations and their corresponding zeta potential in Figure 11. Each black dot represents the exact particle size and zeta potential of the nanoemulsions.

Blood was withdrawn by heart puncture after scarification to verify a complete accumulation of the i.v. injected nanoemulsions from the bloodstream into the organ tissue. With all mean PAREs of the blood below 1%, nearly no nanoemulsion droplets did circulate in the bloodstream anymore for all three negatively surface-charged nanoemulsion formulations, similar to the four previously investigated neutral surface-charged nanoemulsions [24]. The nanoemulsion formulations either accumulated in the organ tissue within 24 h after i.v. application or were already excreted hepatically.

The highest accumulation for all three negatively surface-charged nanoemulsions was detected in the liver tissue, since the liver takes up to 80–90% of the RES function [52,55,56]. The PAREs decreased significantly with increasing particle size, from 51.8% mean PARE for the negative NE50 (zeta potential of −28.8 mV) down to 42.1% mean PARE for the negative NE150 (zeta potential of −20.2 mV), as shown in Figure 11a. Thereby, the significance was prominent between all three particle sizes. In comparison with the previously investigated neutral surface-charged nanoemulsions with zeta potentials between −2.0 and −3.2 mV, the hepatic accumulation was even more dependent on the particle size, decreasing significantly from 66.2% mean PARE for the smallest nanoemulsion NE25 down to 40.3% mean PARE for the NE150 in adult mice [24]. MWU of the similar-sized nanoemulsions with neutral and negative surface charge revealed a significant reduction of the PAREs by the incorporation of the anionic PG into the stabilizing layer of the NE50. For the nanoemulsions NE100 and NE150, the negative surface charge did not change the PAREs significantly in comparison to the neutral surface charge.

Containing reticular cells, the spleen and bone marrow are further parts of the RES as well [57]. The particle size of the negatively surface-charged nanoemulsions had no significant effect on the PARE values and thus on the accumulation in the spleen with mean PAREs between 8.2 and 9.9%, as shown in Figure 11b. In contrast, the particle size of the previously investigated neutral surface-charged nanoemulsions had significant effects on the PARE values with mean PAREs between 6.7 and 12.9%, with their maximum peak at a particle size of 100 nm [24]. Except for the pair of the neutral NE50 and NE100, the significance was prominent between all other pairs of the four different particle sizes. The surface charge of similar-sized nanoemulsions had significant effects on the PARE values. The accumulation in the excised femur, knee, and tibia showed no dependency on the particle size for the negatively surface-charged nanoemulsions with mean PAREs between 2.9 to 3.3% (not depicted as logarithmic contour plot). The previously investigated neutral surface-charged nanoemulsions had mean PAREs between 3.3 and 4.6% [24]. Statistical analysis by MWU between the pairs of similar-sized nanoemulsions with the negative and neutral surface charges revealed that the negative NE50 and negative NE100 accumulated significantly lower in the femur, knee, and tibia than their neutral counterparts. No significant effect of the surface charge was found between the accumulation of the neutral NE150 and negative NE150. Therefore, the detected accumulation in the excised femur, knee, and tibia was probably caused by the uptake of the nanoemulsions into the bone marrow as part of the RES.

The previously investigated neutral surface-charged nanoemulsions revealed a particle size dependency for the accumulation in the reproductive female sex organs uteri and ovaries, especially during their fertile lifetime period in adult mice with up to 9.8% mean PARE for the neutral NE150. The PAREs of the neutral NE150 were significantly higher than the PAREs of the smaller-sized neutral surface-charged nanoemulsions, and therefore may be suitable for the treatment of ovarian diseases or dysfunctions [24]. In contrast to that, all three negatively surface-charged nanoemulsions accumulated independently of their particle size in the female sex organs with nearly identically mean PAREs between 4.8 and 5.1%, as shown in Figure 11c. There was no significant difference in the PAREs caused by the nanoemulsion particle size. Consequently, unintended accumulation of these nanoemulsions in the female sex organs remained at low amounts through the implementation of innovative particle design by incorporation of the anionic phospholipid PG into the stabilizing MHS layer. As a result, the possibility of side effects in the ovaries can effectively be reduced while maintaining a tunable biodistribution profile by adjusting the particle size of the negatively surface-charged nanoemulsions.

Dependent on the particle size, the negatively surface-charged nanoemulsions accumulated in the heart and kidneys, as shown in Figure 11d,f. The mean PARE increased with increasing particle size of the nanoemulsions from 3.2% up to 8.1% in the heart and from 3.3% up to 5.0% in the kidneys. Pairwise statistical analysis by MWU revealed that the increase of the PAREs in the heart was significant from the negative NE50 to the negative NE100 and negative NE150. The PAREs between the negative NE100 and negative NE150 were not significant, but the ongoing increasing mean PAREs suggested that the cardiac accumulation further increased with increasing particle size. In the kidneys, the PAREs of the negative NE50 were significantly lower than the similarly high PAREs of the negative NE100 and negative NE150. This accumulation behavior was observed as well but less intensively with the previously investigated neutral surface-charged nanoemulsions: independently of aging for the accumulation in the heart and in the case of the kidney, accumulation increasing with the aging of the mice [24]. Therefore, the surface charge may influence the cardiac accumulation, since the PAREs of the negative NE100 were significantly higher than the neutral NE100. Although the PAREs of the negative NE50 and negative NE150 were not significantly higher than the PAREs of the similar-sized neutral surface-charged nanoemulsions, their higher mean PAREs suggested an increased cardiac accumulation through the modified negative surface of the nanoemulsion droplets. For the accumulation in the kidneys, no significant effects of the surface charge were detected. The effect of particle size on the accumulation in the heart and kidneys was found as well with other i.v. injected nanoparticular formulations: gold nanoparticles showed an increasing particle size-dependent heart accumulation in pregnant and non-pregnant mice [58], as well as lipid nanocapsules for both heart and kidney accumulation in mice aged 6–8 weeks [59].

As shown in Figure 11e,g, mean PARES were detected with up to 7.7% in the stomach with the negative NE150 significantly higher than the smaller-sized negatively surface-charged nanoemulsions and up to 4.3% in the caecum with no significant effect of the particle size for the negatively surface-charged nanoemulsions, but significant for the neutral surface-charged nanoemulsions. This increased fluorescence signal might derive partially from coprophagy, which was found to begin in the mice age of 2.5 weeks, peak at age five to six weeks with 13 pellets per day, and decreases gradually down to 2.1 pellets daily at an age of 78 weeks [60]. Hence, the adult mice aged 15–30 weeks might reuptake the excreted fluorescent dye DiR into their gastrointestinal tract, which probably already left the body hepatically. Similar observations were made with the previously investigated neutral surface-charged nanoemulsions: a substantial fluorescence signal in the stomach and caecum were found to increase with aging in adult and senescent mice at an age <12 weeks, but not in juvenile mice at an age of three to four weeks, although coprophagy was occurring. Therefore, it was found that the increasing fluorescence signal with aging after puberty stood in contrast to the decreasing coprophagic activity [24]. Consequently, the detected fluorescence signal of the negatively surface-charged nanoemulsions in the stomach and caecum derived likely not only by coprophagy but also through aging effects after puberty, where the RES capacity was found to decline [51,52,53]. This might lead to the more widely distributed accumulation of the negatively surface-charged nanoemulsions in the adult mice across the described other non-RES organs including the uterus, ovaries, heart, stomach, kidneys, and caecum, similar to the previous investigated neutral surface-charged nanoemulsions [24].

All other excised organs (lungs, s.c. fat, pancreas and fat, duodenum, skin, bladder, brain, thigh muscle, and colon) had mean PAREs below 3.0% (in declining order), and thus had just a marginal accumulation of the negatively surface-charged nanoemulsions similar to the previously investigated neutral surface-charged nanoemulsions [24].

Figure 12 shows ex vivo fluorescent images of the seven excised organs with the highest negatively surface-charged nanoemulsions accumulations of the representative mice (one for each nanoemulsion according to Equation (10)). The previously discussed courses of the mean PAREs were confirmed by these ex vivo images, in which the radiant efficiency was mostly evenly distributed over the organs with no distinctive hot spots, except for the female sex organs. The hot spots with high radiant efficiency in the female sex organs were detected in the ovaries for both negatively and the previously investigated neutral surface-charged nanoemulsions [24], as well as many other different lipid or polymer nanoparticulate DDS by fluorescent and nuclear imaging [4,7,8,11,12,13].

## 4. Conclusions

Particle engineering of nanoemulsions was successfully applied by the incorporation of the anionic phospholipid PG into the stabilizing MHS layer of the nanoemulsion droplets. Therewith, the tunable particle properties’ particle size and negative surface charge were achieved. The negatively surface-charged nanoemulsions showed a greatly reduced cellular toxicity on 3T3 and NHDF fibroblasts in comparison to previously investigated neutral surface-charged nanoemulsions of the same particle sizes without any incorporated phospholipids in their stabilizing layer.

The tailor-made nanoemulsion properties’ surface charge and particle size greatly influenced the biodistribution profile in the following pattern: by far the highest accumulation was expectedly found in the liver, which significantly decreased with increasing particle size for both negatively and neutral surface-charged nanoemulsions. While no significant effect of the particle size was detected for the accumulation in the spleen with the negatively surface-charged nanoemulsions, the particle size of the neutral surface-charged nanoemulsions showed significant effects with its maximal accumulation of the neutral NE100 at 13% mean PARE. The accumulation in the heart significantly increased with the increasing of the particle size of both negatively and neutral surface-charged nanoemulsions. As a result, the negatively surface-charged nanoemulsions had higher degrees of cardiac accumulation than their similar-sized neutral surface-charged counterparts. Furthermore, the particle size significantly affected the accumulation in the stomach with their maxima at the biggest particle size for both negatively and neutral surface-charged nanoemulsions. Interestingly, the particle size of the negatively surface-charged nanoemulsions did not affect the accumulation in the female sex organs, since all three different-sized formulations accumulated nearly identically at mean PAREs of 5%. In contrast, the neutral surface-charged nanoemulsions accumulated dependent on their particle size, where the highest accumulation of the neutral NE150 with 10% mean PARE significantly differed from the smaller-sized formulations. The mean PAREs of all other organs were below 5% for all formulations.

Consequently, innovative particle engineering of these nanoemulsions proved to be a powerful tool to alter the biodistribution profile and thereby to modify the accumulation in different organs. Of special interest was the accumulation in the female sex organs with hot spots in the ovaries: the accumulation of the nanoemulsions remained at constant low amounts through the implemented negative surface charge for all three tested particle sizes. Thus, possible side effects through unintended drug delivery or nanotoxicology of the carrier itself can effectively be reduced while maintaining a tunable biodistribution profile by adjusting the particle size of the negatively surface-charged nanoemulsions. In case a drug delivery into the ovarian tissue is desired, using the previously investigated neutral surface-charged nanoemulsions with the biggest particle size of 150 nm in diameter might increase the treatment success of ovarian dysfunctions or diseases, like primary ovarian insufficiency or ovarian cancer, since this formulation showed the highest ovarian accumulation in adult mice [24].

## Figures and Tables

**Figure 1 pharmaceutics-14-00301-f001:**
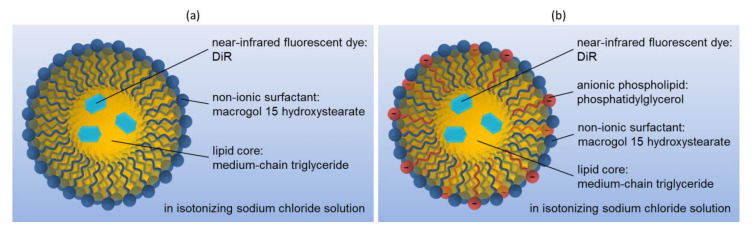
Composition of the nanoemulsions for the in vivo biodistribution studies: (**a**) previously in vivo investigated neutral surface-charged nanoemulsions [24] and (**b**) surface modified negatively surface-charged nanoemulsions through the integration of the anionic phospholipid PG into the stabilizing layer.

**Figure 2 pharmaceutics-14-00301-f002:**
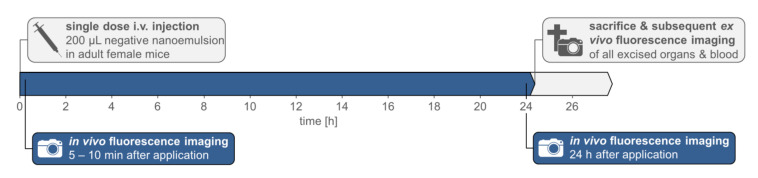
Experimental procedure of the animal trials.

**Figure 3 pharmaceutics-14-00301-f003:**
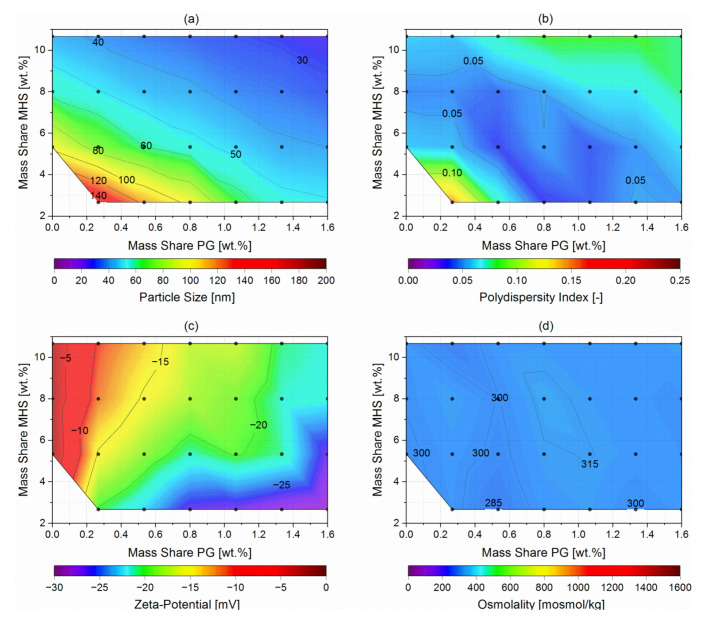
Contour plots showing (**a**) the resulting particle diameter, (**b**) the PDI, (**c**) the zeta potential in 0.1× PBS at pH 7.4, and (**d**) the osmolality of the screening trials depending on the MHS against the PG mass shares at a constant MCT mass share of 8 wt.%, in which each dot ● represents a conducted experiment.

**Figure 4 pharmaceutics-14-00301-f004:**
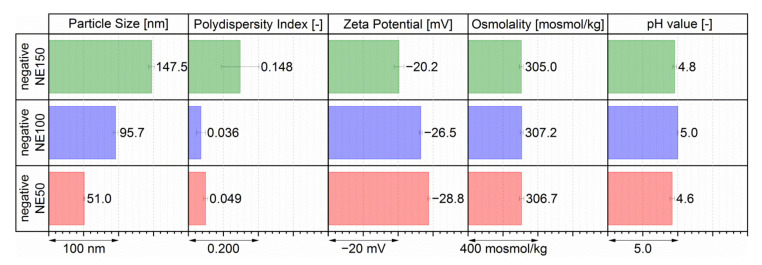
Physicochemical properties of the three chosen isotonic, negatively surface-charged nanoemulsions for the animal trials, determined by three individual produced batches.

**Figure 5 pharmaceutics-14-00301-f005:**
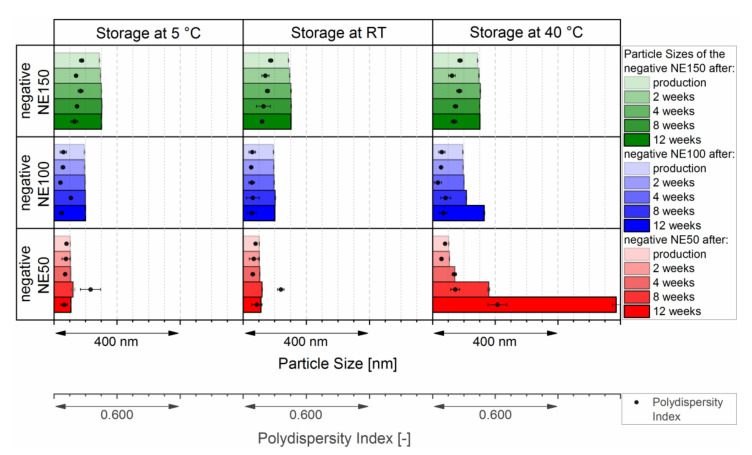
Long-term stability of the three negatively surface-charged nanoemulsions at the recommended storage conditions according to the ICH guidelines Q1A of 5 ± 3 °C, room temperature (RT), and 40 ± 2 °C.

**Figure 6 pharmaceutics-14-00301-f006:**
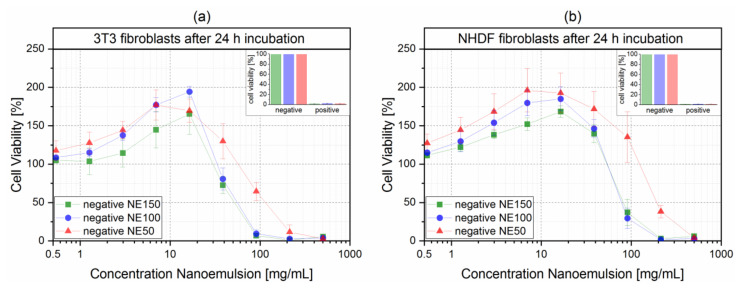
Dose-response curves of the cell viability over the total mass nanoemulsion (DiR loaded MCT, PG, MHS + aqueous phase) per lmL cell culture media on (**a**) 3T3 and (**b**) NHDF fibroblasts after 24 h incubation with the three negatively surface-charged nanoemulsions, determined by three individual incubated batches (eight replicates per run); the inlet graphs show the corresponding negative and positive controls with the untreated or TritonTM X-100 treated cells, respectively.

**Figure 7 pharmaceutics-14-00301-f007:**
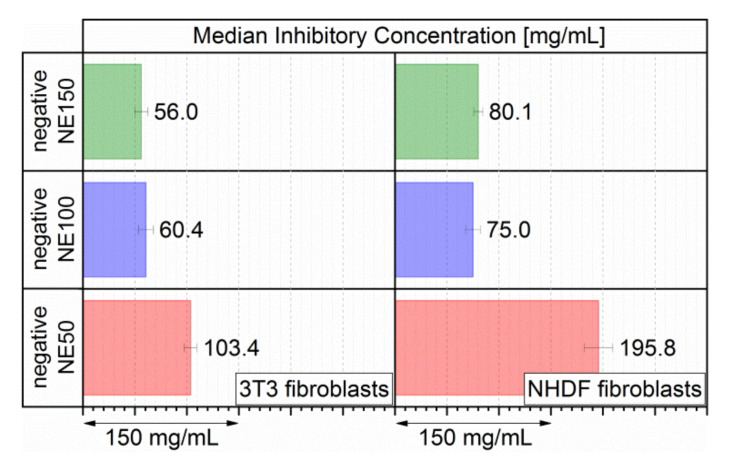
IC50 as the total mass of the three negatively surface-charged nanoemulsions (DiR loaded MCT, PG, MHS + aqueous phase) per mL cell culture media on 3T3 and NHDF fibroblasts after 24 h incubation, determined by three individual incubated batches (eight replicates per run).

**Figure 8 pharmaceutics-14-00301-f008:**
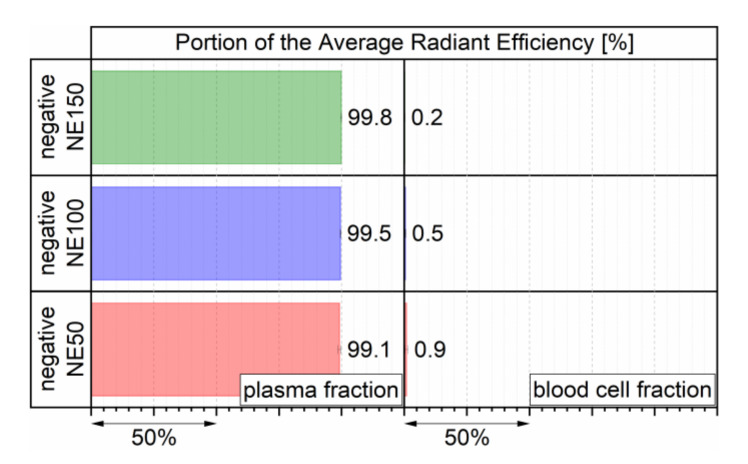
PARE in plasma and blood cell fractions after 4 h of incubation with the three negatively surface-charged nanoemulsions, determined by three individual incubated batches.

**Figure 9 pharmaceutics-14-00301-f009:**
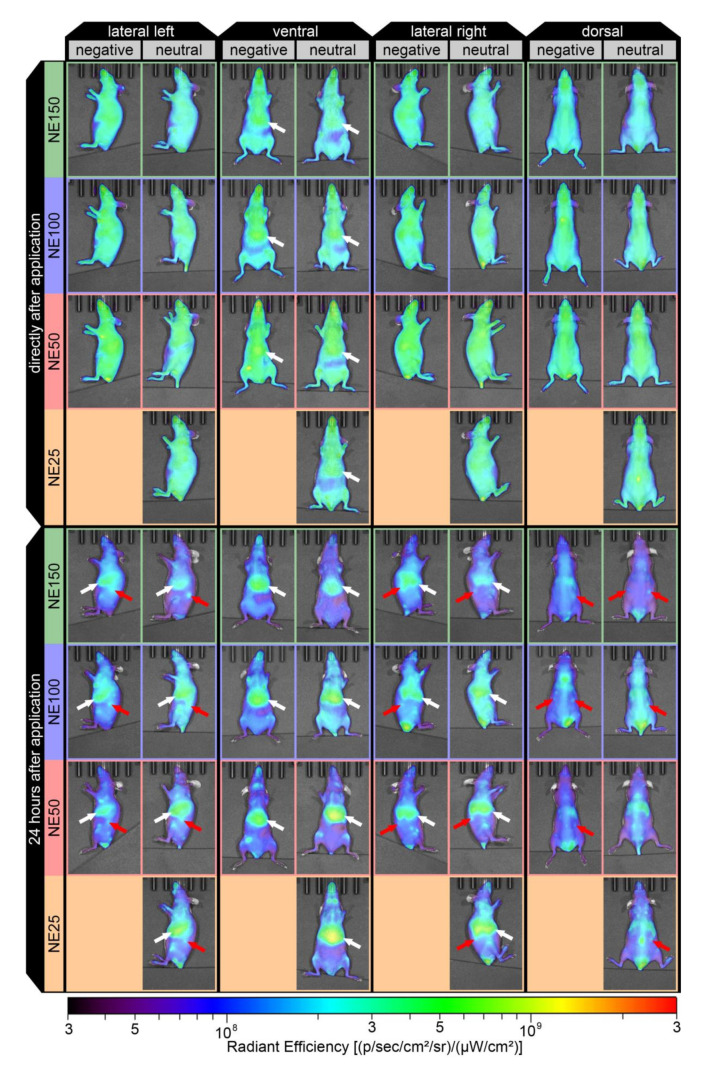
Noninvasive in vivo fluorescence images of the representative mice directly after (5–10 min) and 24 h after i.v. injection with the three negatively surface-charged nanoemulsions and an additional four previously investigated neutral surface-charged nanoemulsions [24].

**Figure 10 pharmaceutics-14-00301-f010:**
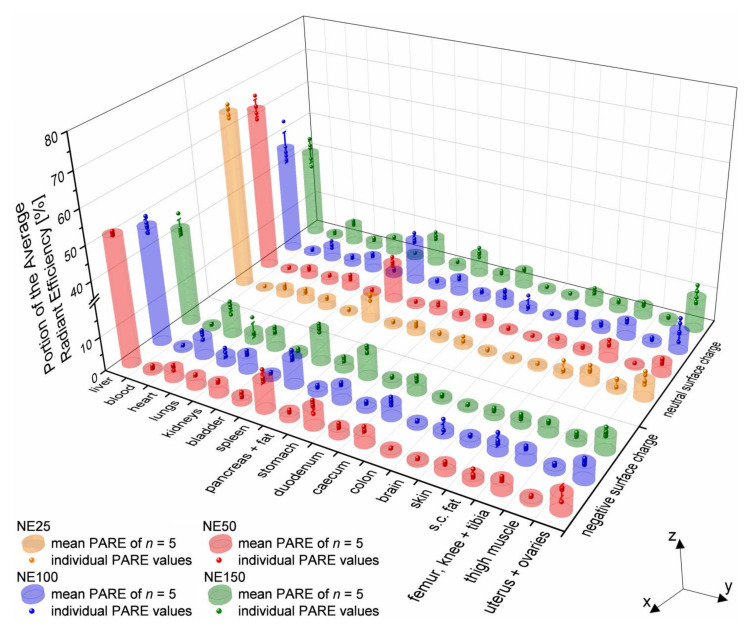
3D bar and scatter chart of the PARE of the ex vivo excised organs and blood, 24 h after i.v. injection of the three negatively surface-charged nanoemulsions and additionally four previously investigated neutral surface-charged nanoemulsions [24].

**Figure 11 pharmaceutics-14-00301-f011:**
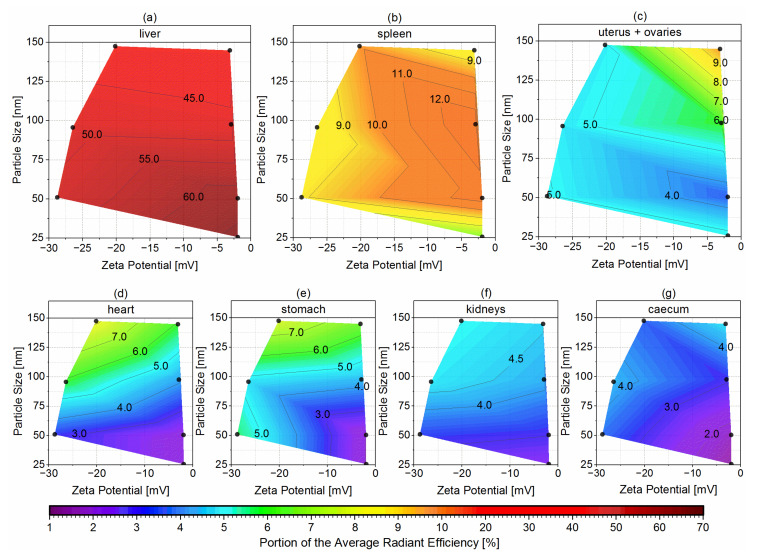
Logarithmic contour plots of the mean PARE for the organs with the highest accumulations ((**a**) liver, (**b**) spleen, (**c**) uterus + ovaries, (**d**) heart, (**e**) stomach, (**f**) kidneys, and (**g**) caecum) plotted against the particle size and the zeta potential of the three i.v. injected negatively surface-charged nanoemulsions and the additional four previously investigated neutral surface-charged nanoemulsions [24], displayed as ●.

**Figure 12 pharmaceutics-14-00301-f012:**
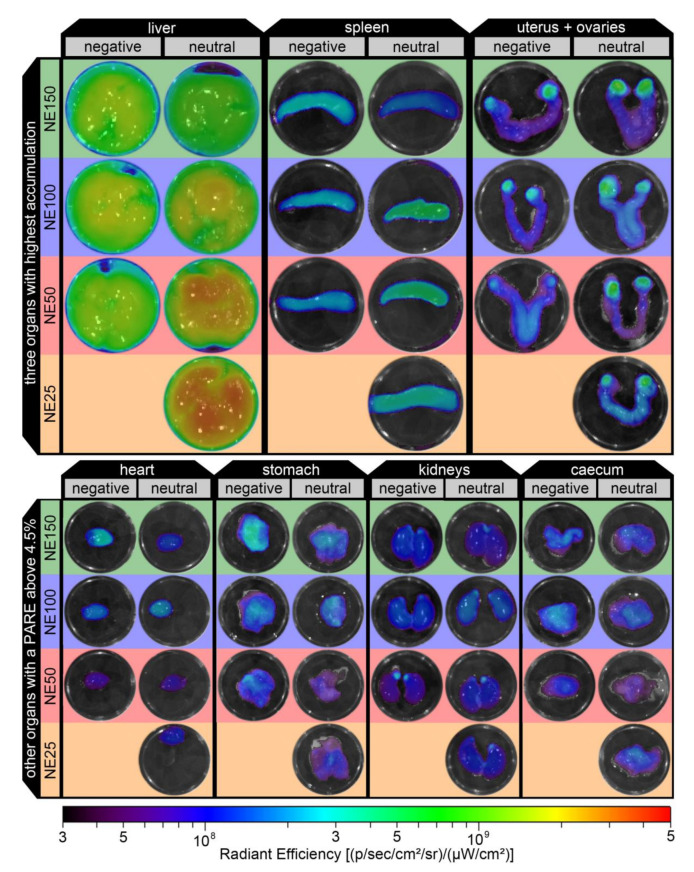
Ex vivo fluorescence images of the excised organs with the highest accumulation (liver, spleen, uterus +ovaries, heart, stomach, kidneys, and caecum) of the representative mice with the three i.v. injected negatively surface-charged nanoemulsions and additionally four previously investigated neutral surface-charged nanoemulsions [24].

**Table 1 pharmaceutics-14-00301-t001:** Composition of the isotonic, negatively surface-charged nanoemulsions for the animal trials.

Compounds in wt.%	Negative NE50 ^1^	Negative NE100 ^1^	Negative NE150 ^1^
DiR loaded MCT ^2^	8.00	8.00	8.00
MHS	2.67	2.67	2.67
PG	1.60	0.80	0.27
NaCl solution	14.40 ^3^	15.20 ^4^	15.73 ^5^
ice-cold water	73.33	73.33	73.33

^1^ NE50, NE100, and NE150 refer to the nanoemulsion particle size of 50, 100, and 150 nm, respectively; ^2^ DiR loaded MCT at a concentration of 0.1 mg/g; NaCl solutions at salinities of ^3^ 5.05 wt.%, ^4^ 4.83 wt.%, and ^5^ 4.69 wt.%.

## Data Availability

Not applicable.

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
