# Peer review of "Particle Engineering of Innovative Nanoemulsion Designs to Modify the Accumulation in Female Sex Organs by Particle Size and Surface Charge"

_pharmaceutics, 2022, doi:10.3390/pharmaceutics14020301_

Round 1
Reviewer 1 Report
Bussmann and colleagues described a modified nanoemulsion system based on their previous research, loaded with a contrast agent and investigated for its potential use intravenously. Indeed, intravenous administration of nanoemulsion meets plenty of limitations and therefore, surface modification can reveal promising results. This approach is up-to-date, as until now the potent use of nanoemulsions modified with PG has not been extensively examined for intravenous drug delivery, especially through in vivo and ex vivo approaches. From this point of view, this research reveals interesting data, by confirming the necessity of nanoemulsion’s physicochemical modulation. I recommend publication of the research article after some clarifications and comments that are essential to my opinion.
- In section 2.2 (lines 165-170) the protocol of particle size measurement and zeta potential analysis reports that all nanoemulsion systems were assessed at room temperature. Nanoemulsions’ stability can be affected by temperature, pH and saline conditions. As these systems are proposed for intravenous delivery, it would be interesting to investigate particle size and zeta potential values at 37℃ (average body temperature).
- Bussmann et al. previously conducted stability studies of neutral nanoemulsions as described in reference [23]. However, for reasons of comparison, it would be interesting to investigate nanoemulsions’ stability when modified with phosphatidylglycerol.
- Concerning in vitro toxicity (lines 176-178), Bussmann and colleagues chose human dermal fibroblasts and mouse embryonic fibroblasts. Indeed, fibroblasts are in general sensitive cell lines and are frequently used in toxicity experiments. However, this study investigates the accumulation of nanoemulsions in female sex organs, so it would be interesting to evaluate their cytotoxicity in cell lines originated from sex organs (eg. ovarian or uterus cell lines) if this is applicable.
- In in vitro toxicity experiment, cells were incubated for 24h after nanoemulsion administration (lines 191-192). Did authors investigate the effect of nanoemulsion administration after 48h or 72h of incubation, as the majority of cell lines duplicate after the first 24h? Is this the case for the selected cell lines?
- In lines 357-358 the authors referred to growth-promoting effects of the nanoemulsions tested. A possible explanation of this phenomenon is missing in this section.
- In lines 594-595 authors concluded that the size of modified nanodroplets does not affect accumulation. This phenomenon is observed due to all results obtained through nanoemulsion’s modification with phosphatidylglycerol. It is not clear why does the negative charge exhibits different accumulation degree compared to neutral.
Reviewer 2 Report
Dear Lucas and Busmann
This report investigates the influence of particle negative surface in its accumulation in female sex organs overcame particle size dependence. The experimentation was done in a systematic manner and the report was written fairly well. The manuscript can be further improved after minor revisions suggested below:
- In the abstract (p.1) it was not clear the objective and conclusion of the work.
- 10; Line.369 “Thereby, a formation of needle-like 12 hydroxystearic acid crystals as a metabolic degradation product of MHS likely caused cell death in vitro but is unlikely to occur in vivo because of different transport and metabolic conditions [23].” Are there in vivo studies that confirm this assertion? Show these studies.
- Page 12, line 447 and page13, Figure 8: the NE25 formulation was was cited in the manuscript without prior information. was a particle developed in another work? what physicochemical characteristics?
- Page 14, Figure 9: insert statistical analyses into the results shown in the figure.
- Page 15, line 504-510; page 16, line 511-512. Show statistical analysis. Is the decrease in the PARE mean for negative particles significant? and for neutral particles? Was there a significant difference when comparing a negative particle with a neutral particle? For the 150 nm particle, the neutral and negatively charged particles showed the same mean PARE.
- The results show an accumulation of negative particle size-dependent nanoemulsion in some organs and charge-dependent in the ovary. Is there a reason for this variation to justify the work title?
- The conclusion is not very clear and must be in line with the objective of the work, which was not very clear. Is the main objective to reduce side effects or increase therapeutic effectiveness? negative nanoemulsion seems to reduce side effects while neutral nanoemulsion seems to improve therapeutic efficacy when it presents larger globule sizes.
Reviewer 3 Report
The authors describe the production and the physicochemical characterization of NE with defined size intervals by changing parameters of its composition. The authors also successfully showed the reduction in toxicity of NE by incorporating the anionic phospholipid phosphatidylglycerol, and by reducing the wt% of macrogol 15 hydroxystearate in comparison to previous studies; this approach lead to a better delivery system for in vitro application. Authors also show very interesting in vivo and ex-vivo results concerning the target organs of the produced NE and also the degree of NE accumulation in those organs.
The introduction is well written and describes the previous research needed to understand the aims of the work. The whole manuscript is well written and results are supported by well-designed and comprehensive figures and tables. The study is very interesting, is very well designed, the data is well presented and discussed, however some points need to be clarified /correct before its publication.
Next I will point, sequentially, all points that raised some doubts or that need correction.
Keywords: maximum 10 keywords are accepted, please see https://www.mdpi.com/journal/pharmaceutics/instructions
Line 140. Please, include in the text the definition of all terms, for all the listed equations; this will allow the reader to better understand/understand the meaning and purpose of each equation.
Table 1. Please include the meaning of NE50, NE100 and NE150, as it was not defined before. Or define it at lines 150-151.
Line 176-178, please add the origin of the cells (of NHDF and of 3T3 cells).
Line 212, please replace “chapter 3.7” by “section 2.7”.
Line 214, please replace “chapter 3.8” by “section 2.8”.
Line 215, please renumber the section (3.6 should be 2.6)
Data in Figure 5 show that low concentrations of the NE induce growth promoting effects; with NE100 doubling the cell population (Figure 5a) at 16.4 mg/mL. In the authors opinion, what is promoting this cell population duplication? Which would be the consequences in vivo? And, as authors applied the resazurin on top of test solutions, I would like to know if there is any type of interference between resazurin fluorescence and the NE, did authors tested that?
As particles in suspension may deviate the emitted light, or may obstruct light, or interact with resazunin fluorescence, did authors considered the replacement of test solutions by resazunin solution instead of adding resazunin directly on top of test solutions?
I think Figure S3 would benefit if the authors included the respective negative cell controls, cells exposed only to the culture media. This would also serve to prove the data in figure 5 (increased cell proliferation).
Line 386-387. Is the sentence “… less rapid decrease of cell viability with increasing nanoemulsion concentration” correct? From the figure 5 and from the IC50, the higher concentration gave a faster decrease in cell viability which corroborates with a lower IC50. Please revise.
466-67, please rephrase.
Figure 10. Values in figure have a comma as separator, shouldn’t it be a dot?
Round 2
Reviewer 3 Report
The authors have answered all the posed questions. Thank you.
The authors have modified/corrected/clarified the suggested points, as requested.
I would like to suggest a few minor corrections.
Minor corrections,
Table 1, first row, I do not see the need of adding the word “negative”, as it is in the table legend. You could add to the legend the following: “NE50, NE100 and NE150 refer to nanoemulsion particle size of 50, 100 and 150 nm, respectively”
Or place this sentence as Table footnote to indicate the meaning of NE50…
Line 262. Replace “i.v.” by “intravenously (i.v.)”
Line 267, format the chemical formula of oxygen.
In this subsection two different notations for litre are used “L” and “l”, I recommend to use only one, and I suggest using the L (capitalized), please revise the whole document (use the same notation in the whole manuscript)
